# Remaining Useful Lifetime Prediction Based on Extended Kalman Particle Filter for Power SiC MOSFETs

**DOI:** 10.3390/mi14040836

**Published:** 2023-04-12

**Authors:** Wei Wu, Yongqian Gu, Mingkang Yu, Chongbing Gao, Yong Chen

**Affiliations:** School of Automation Engineering, University of Electronic Science and Technology of China, Chengdu 611731, China

**Keywords:** SiC MOSFETs, RUL estimation, Extended Kalman Particle Filter, reliability

## Abstract

Nowadays, the performance of silicon-based devices is almost approaching the physical limit of their materials, which have difficulty meeting the needs of modern high-power applications. The SiC MOSFET, as one of the important third-generation wide bandgap power semiconductor devices, has received extensive attention. However, numerous specific reliability issues exist for SiC MOSFETs, such as bias temperature instability, threshold voltage drift, and reduced short-circuit robustness. The remaining useful life (RUL) prediction of SiC MOSFETs has become the focus of device reliability research. In this paper, a RUL estimation method using the Extended Kalman Particle Filter (EPF) based on an on-state voltage degradation model for SiC MOSFETs is proposed. A new power cycling test platform is designed to monitor the on-state voltage of SiC MOSFETs used as the failure precursor. The experimental results show that the RUL prediction error decreases from 20.5% of the traditional Particle Filter algorithm (PF) algorithm to 11.5% of EPF with 40% data input. The life prediction accuracy is therefore improved by about 10%.

## 1. Introduction

Power electronic systems play essential roles in electrical energy transmission and conversion in many applications [1,2,3]. Semiconductor power devices, such as metal-oxide field-effect transistors (MOSFETs) and insulated gate bipolar transistors (IGBTs), are the nucleus devices of power electronic systems, whose failures can cause system losses [4]. However, long maintenance times can increase the operating cost and reduce the efficiency of the entire system. It is very important to research the health statuses of power devices to improve the operational reliability of power electronic systems [5].

Reliability research is mainly divided into two directions, namely condition monitoring and life prediction [6]. Condition monitoring is used to evaluate the current health status while the power device is still working. If the monitored data deviate from the normal health value, the degree of deviation from the normal state should be evaluated [7]. Life prediction is used to assess the health state of the power device in the future and estimate the remaining useful life (RUL) under current operating conditions. If the RUL of the power device is known, the occurrence of catastrophic accidents will be avoided by taking some early warning measures [8]. Therefore, life prediction is a significant method of reliability evaluation. The life prediction methods mainly include the failure model method and the data-driven method. In recent years, many life prediction methods have been proposed. In [9], a prediction method based on Gaussian process regression (GPR) and Kalman filtering (KF) for a battery’s end-of-life has been proposed. Compared to the traditional particle filter (PF)-based end-of-life prediction framework, the framework proposed in [9] has higher prediction accuracy and smaller range of prediction uncertainty. Mo et al. developed a novel RUL prediction method for lithium-ion batteries based on PF combined with Kalman filtering and particle swarm optimization, which not only improves the precision over standard PF but also overcomes the particle degradation due to particle resampling [10]. L. Cui et al. proposed a novel method for RUL prediction of rolling element bearings based on time-varying Kalman filters, which can conquer the complexity of the bearing degradation process [11]. The degradation process of the bearing was divided into the normal working stage and accelerated degradation stage. A model switching threshold of 5% is proposed to judge the stage of bearing. C. Zhang et al. developed a novel hybrid approach to forecasting battery future capacity and RUL by combining the improved variational modal decomposition (VMD), particle filter (PF), and Gaussian process regression (GPR) [12]. The proposed approach offers wide generality and reduced errors, which have indicated a better prediction performance compared with the individual PF, solo GPR, single VMD-PF, and separate VMD-GPR algorithms. In [13], a fusion neural network model is developed to predict the lithium-ion battery capacity and RUL by combining a broad learning system (BLS) algorithm and long short-term memory neural network (LSTM NN) which can guarantee the precision of the lithium-ion battery capacity and RUL prediction with less training. C. Zhang et al. also published an incremental capacity analysis (ICA) and improved broad learning system (BLS) network-based state-of-health (SOH) estimation technology for lithium-ion batteries that can effectively evaluate the SOH with strong robustness as well as stability to the degradation and disturbance of in-service and retired lithium-ion batteries [14]. Many other life prediction methods for MOSFETs have been proposed [15,16,17,18].

Nowadays, the performance of silicon-based devices is approaching the physical limit of their materials. It is difficult to meet the requirements of modern high-power applications. The third generation of wide bandgap semiconductor materials headed by silicon carbide can well meet the conditions of high temperature and high pressure and will gradually replace silicon materials in high-power applications. Some parameters of Si and SiC materials are compared in Table 1.

SiC MOSFET, as an important third generation wide bandgap power semiconductor device, has numerous specific reliability problems, such as bias temperature instability, threshold voltage drift, and reduced short-circuit robustness, which greatly reduce the reliability of the device [19]. Therefore, the life prediction of SiC MOSFETs is of importance and has become a hot research topic. Several life prediction methods for SiC MOSFETs have been proposed. L. Ceccarelli et al. proposed a fast lifetime prediction strategy for the bond-wire interconnections of SiC MOSFETs. The model of accumulated damage of module under thermal mechanical stress is built, which can be used to evaluate the health state of the power module [20]. A 2D finite element model has been developed, by B. Hu et al., to evaluate the stress performance and lifetime of the solder layer. The lifetime of SiC MOSFETs can be quantified by using the validated stress-based model [21]. S. Zhao et al. built a composite failure precursor formulated by taking full advantage of potential failure precursors of SiC MOSFETs. The genetic programming method is utilized to the nonlinear fusion of the failure precursors. This approach can improve the prediction performance [22]. S. Zhao et al. proposed a health state estimation and remaining useful life prediction method for SiC MOSFETs based on noisy and aperiodic degradation measurements. A stochastic expectation-maximization algorithm is applied to estimate the model parameters. The error of RUL prediction is less than 16%, and this method is robust under various noise levels [23]. Although some life prediction methods have been proposed, there are few researches on the life prediction for SiC MOSFETs, especially based on the PF algorithms [19].

In this paper, a remaining useful life estimation method for SiC MOSFETs using the Extended Kalman Particle Filter (EPF) algorithm based on on-state voltage (*V_ds,on_*) degradation model is proposed. Three different *V_ds,on_* degradation empirical models are also discussed. A new power cycling test platform is designed to monitor the *V_ds,on_* of SiC MOSFETs used as the failure precursor. The EPF algorithm overcomes the particle degradation of the PF algorithm by fusing EKF, which provides a reasonable recommended density distribution for sampling particles. Compared with the PF life prediction method, the EPF method can further improve the accuracy of remaining useful life estimation.

The logical structure of this paper is as follows. Section 2 introduces the failure precursor of SiC MOSFETs and the power cycle test platform. Section 3 introduces the Particle Filter algorithm and its improved algorithm. In Section 4, the RUL estimation results based on the EPF algorithm are presented and compared with that based on the PF algorithm. Section 5 will give the conclusion.

## 2. SiC MOSFET Failure Precursor and Test Setup

### 2.1. Experimental Setup

The chip of the power device should be packaged in order to protect the chip and dissipate heat. There are many kinds of materials in the SiC MOSFETs, such as copper, aluminum, ceramic, and silicon carbide. These materials have different coefficients of thermal expansion (CTE). The heat generated during the on-state of SiC MOSFET will be transferred from the chip to other materials inside the module which can produce the thermomechanical stress. The deformation and degradation of material caused by these stresses will eventually lead to device failure. The main failure mechanisms of power devices are package-level failures which mainly include solder layer fatigue and bond wire failure [24]. In order to simulate the degradation process of SiC MOSFETs in the actual system with a relatively short time, the accelerated degradation of SiC MOSFETs is implemented by the power cycling test, which causes package-level failures [25].

The schematic diagrams of the experimental setup and the power cycling test platform are shown in Figure 1 and Figure 2, respectively. The device under test (DUT) uses the discrete semiconductor device of SiC MOSFETs. In this paper, the SiC MOSFET G3R350MT12D is selected, whose operating temperature range is −55 °C to 175 °C [26]. A platinum thermal temperature sensor PT1000 is attached to the heat dissipation surface of the SiC MOSFETs which is used to measure the case temperature (*T_c_*) of the SiC MOSFETs. The temperature data processed by MAX31865 is transferred to the PC through SPI communication. IR25750LPBF is utilized to measure *V_ds,on_* during the on-state of SiC MOSFETs. The control board using a STM32F4 series chip module controls the switch of SiC MOSFETs and stores the data of *V_ds,on_*. The power load resistor R is set to 5 Ω to limit current. The power supply DC voltage is set to 40 V.

The experimental procedure and the case temperature swing of SiC MOSFET during the power cycling test are shown in Figure 3 and Figure 4, respectively. The minimum case temperature *T_min_* and maximum case temperature *T_max_* in the power cycling test are set as *T_min_* = 30 °C and *T_max_* = 190 °C, respectively, which apply strong stresses on the DUT to accelerate degradation. *T_m_* is the average case temperature, which is set as *T_m_* = 110 °C. The on-state current *I_C_* is set to about 8 A, which is also slightly above the maximum rating current of DUT. The *V_ds,on,th_* is the failure threshold value of the *V_ds,on_*. The gate voltage *V_GS_* is set as 15 V. When *V_GS_* is 15 V, the DUT is turned on and the case temperature rises to *T_max_*. Then, *V_GS_* drops to 0 and the device is turned off until it is turned on again when the DUT temperature drops to *T_min_*. *V_ds,on_* is measured during the on-state of SiC MOSFET when *T_c_* = *T_m_* to reduce the impact of temperature. The main parameters of experimental setup are listed in Table 2.

### 2.2. Failure Precursor

The package-level failures can be reflected by the failure precursors, which are the external electrical parameters of power devices. Some researchers have used on-state resistance (*R_ds,on_*) as the failure precursor for MOSFETs [15]. The *R_ds,on_* is calculated by dividing the on-state voltage by the on-state current; however, measuring the voltage and current at same time will introduce more noise and errors. SiC MOSFETs have smaller *R_ds,on_* than Si MOSFETs at the same power level. Thus, the measurement accuracy of *R_ds,on_* for SiC MOSFETs is even worse.

In this paper, three different failure precursors of the on-state voltage (*V_ds,on_*), the threshold voltage (*V_th_*), and gate voltage (*V_ge_*) are compared and discussed for SiC MOSFETs. Figure 5 shows the threshold voltage *V_th_* variation with the number of power cycles for different drain currents *I_d_*. The *V_th_* can be given by [27]:(1)$$V_{th}=\phi_{ms}-\frac{Q_{SS}}{C_{ox}}+2\Phi_{FB}+2\frac{\sqrt{qN_{A}\varepsilon_{s}\Phi_{FB}}}{C_{ox}}$$
where *Q_SS_* is the charge at the silicon-silica interface, φ_*ms*_ is the work function between metal and semiconductor, and *N_A_* is the doping concentration of the body region of SiC MOSFETs. The bond wire failure, as the most common failure mode of package-level failures, reduces the overlap area between the polysilicon gate and the active region of the device, which decreases the gate oxide capacitance *C_ox_*. According to Equation (1), the *V_th_* rises. It can be observed that the *V_th_* increases monotonically with the number of power cycles. The change rates of *V_th_* for *I_d_* = 250 μA, *I_d_* = 1 mA, and *I_d_* = 2 mA are 16.8%, 14.7%, and 13.7%, respectively, after 1000 power cycles.

The turn-on transient gate voltage *V_ge_* waveforms recorded by a Tektronix MDO3014 oscilloscope at different numbers of power cycles are shown in Figure 6. The comparison is highlighted in the inset. For the first rising phase of *V_ge_*, it is almost unchanged at the various aging stages. During the phase of the Miller plateau, the plateau duration (*t_GP_*) is reduced with the increase in the number of power cycles. For the second rising phase of *V_ge_*, the slopes of the change in *V_ge_* are nearly identical at different numbers of power cycles. After 1000 power cycles, the *t_GP_* decreased by about 90 ns.

Figure 7 shows the dependence of the variation in *V_ds,on_* monitored by the power cycling test platform on the number of power cycles. The failure threshold is 1.2 times the initial value of *V_ds,on_* [28]. The *V_th_* can be given by:(2)$$V_{ds,on}=R_{bw}I_{d}+V_{F,chip}$$
where *R_bw_* is the resistance of bond wire and *V_F,chip_* is voltage drop of the internal chip. According to Equation (2), the bond wire fatigue increases *R_bw_*, resulting in the rise of *V_ds,on_*. It can be observed that the *V_ds,on_* increases monotonically with the increase of the number of power cycles which rises by 35% from 2.46 V to 3.32 V after 1000 power cycles.

The variation rates and absolute values of three different failure precursors after 1000 power cycles are compared in Table 3. It demonstrates that the *V_ds,on_* and *V_th_* increases, while the *t_GP_* decreases, with the aging of DUT. However, the change in *V_th_* is less pronounced, which is easy to be interfered with by noise. The online monitoring of *V_th_* is difficult to implement because the operation of the power system must be interrupted during the measurement of *V_th_* and a detection signal needs to be injected into the gate electrode. Although the *t_GP_* changes significantly, detecting small absolute values of variation in *t_GP_* from the fast and high-frequency IGBT switching transient is still difficult in practice due to the requirement of a high-precision measurement circuit with an extremely high sampling rate. The change of *V_ds,on_* can reflect the same failure mechanism with *R_ds,on_* [19]. The online measurement of *V_ds,on_* can be implemented and introduces less noise. Therefore, *V_ds,on_* is selected as the failure precursor parameter for SiC MOSFETs in this paper.

## 3. Extended Kalman Particle Filter

### 3.1. The Particle Filter Algorithm

The Particle Filter algorithm collects a group of random samples distributed in the state space to approximate the posterior probability density distribution. It uses the mean of these samples to replace the integral operation in the Bayesian Filter algorithm. Then, the minimum variance estimation of the system state is obtained. These random samples are visually called particles. Particle Filter is a method of approximating the Bayesian Filter algorithm based on the Monte Carlo theory. It is also a kind of sequential importance sampling method. The superiority of the Particle Filter algorithm in nonlinear and non-Gaussian systems determines its wide range of applications. Therefore, the Particle Filter algorithm can show a good performance for tracking and predicting the life of SiC MOSFETs.

The system model for estimating the trajectory of *V_ds,on_* includes the following two parts:(3)$$V_{ds,on,pre,n}=f\left(V_{ds,on,pre,n-1}\right)+w_{n-1}$$
(4)$$V_{ds,on,act,n}=h\left(V_{ds,on,pre,n}\right)+v_{n}$$

Equation (3) is the state transition model and f is the state transition function. Equation (4) is the measurement model and *h* is the measurement function [28]. *V_ds,on,pre,n_* is the predicted value of *V_ds,on_* at time *n*, and *V_ds,on,act,n_* is the measured value of *V_ds,on_* at time *n*. The values *w* and *v* are the process noise and measurement noise of the system, respectively.

Since the measured value contains noise in the system, the Particle Filtering algorithm is used to filter out this noise to obtain the real value. The expected value *E*[*f*(*V_ds,on,pre,n_*)] is the filtered value of *V_ds,on_* at time n, which can be calculated as follows:(5)$$E\left[f\left(V_{ds,on,pre,n}\right)\right]=\int f\left(V_{ds,on,pre,n}\right)p(V_{ds,on,pre,n}|V_{ds,on,act,1:n})dV_{ds,on,pre,n}$$
where *f(V_ds,on,pre,n_)* is the state transition function at time *n*, and *p(V_ds,on,pre,n_|V_ds,on,act,1:n_)* is the posterior probability distribution function at time n calculated by Bayesian Filter algorithm.

Bayesian Filter theory mainly consists of two steps: prediction and update. For the prediction step, *V_ds,on,pre,n_* is estimated by using the Bayesian total probability formula based on the previous measured value as follows:(6)$$\begin{array}{l} p(V_{ds,on,pre,n}|V_{ds,on,act,1:n-1})\\ \;=\int p(V_{ds,on,pre,n},V_{ds,on,pre,n-1}|V_{ds,on,act,1:n-1})dV_{ds,on,pre,n-1}\\ \;=\int p(V_{ds,on,pre,n}|V_{ds,on,pre,n-1},V_{ds,on,act,1:n-1})\\ \;\;\;\;\;\;\times p(V_{ds,on,pre,n-1}|V_{ds,on,act,1:n-1})dV_{ds,on,pre,n-1}\\ \;=\int p(V_{ds,on,pre,n}|V_{ds,on,pre,n-1})\\ \;\;\;\;\;\;\times p(V_{ds,on,pre,n-1}|V_{ds,on,act,1:n-1})dV_{ds,on,pre,n-1} \end{array}$$
where *p*(*V_ds_*_,*on*,*pre*,*n*_*|V_ds_*_,*on*,*act*,1*:n*−1_) is the prior probability distribution function, and where *p*(*V_ds_*_,*on*,*pre,n*−1_*|V_ds_*_,*on*,*act*,1:*n*−1_) is the posterior probability distribution function at time *n* − 1.

For the update step, the posterior probability distribution function *p*(*V_ds_*_,*on*,*pre*,*n*_*|V_ds_*_,*on*,*act*,1:*n*_) is obtained by modifying the prediction step with the measured value at time *n* as follows:(7)$$\begin{array}{l} p(V_{ds,on,pre,n}|V_{ds,on,act,1:n})\\ \;=\frac{1}{Z_{n}}p(V_{ds,on,act,n}|V_{ds,on,pre,n},V_{ds,on,act,1:n-1})\times p(V_{ds,on,pre,n}|V_{ds,on,act,1:n-1})\\ \;=\frac{1}{Z_{n}}p(V_{ds,on,act,n}|V_{ds,on,pre,n})\times p(V_{ds,on,pre,n}|V_{ds,on,act,1:n-1}) \end{array}$$
where *Z_n_* is the normalizing constant which can be calculated as follows:(8)$$\begin{array}{ll} Z_{n} & =p(V_{ds,on,act,n}|V_{ds,on,act,1:n-1})\\ & =\int p(V_{ds,on,act,n}|V_{ds,on,pre,n})\times p(V_{ds,on,pre,n}|V_{ds,on,act,1:n-1})\;dV_{ds,on,pre,n} \end{array}$$

The trouble is that the exact form of the posterior probability distribution function *p*(*V_ds_*_,*on*,*pre*,*n*_*|V_ds_*_,*on*,*act*,1:*n*_) is unknown. It is still unclear how to sample a bunch of particles from this posterior probability distribution. At this time, importance sampling should be introduced to solve this problem. The importance sampling method is to find a reference distribution function to sample particles [29]. Thus, Equation (5) can be rewritten as follows:(9)$$\begin{array}{ll} E\left[f\left(V_{ds,on,pre,n}\right)\right] & =\int f\left(V_{ds,on,pre,n}\right)\frac{p(V_{ds,on,pre,n}|V_{ds,on,act,1:n})}{q(V_{ds,on,pre,n}|V_{ds,on,act,1:n})}\\ & \times q(V_{ds,on,pre,n}|V_{ds,on,act,1:n})dV_{ds,on,pre,n} \end{array}$$
where *p*(*V_ds_*_,*on*,*pre*,*n*_*|V_ds_*_,*on*,*act*,1:*n*_) is a known and easily sampled reference distribution.

The integral operation part of Equation (9) is still difficult to calculate. To solve this problem, the Monte Carlo method is introduced into the Bayesian Filter algorithm. It uses a set of particles to approximate the posterior probability. Then, the average of these particles is used to replace the calculation of the integral part. Therefore, the reference distribution function *p*(*V_ds_*_,*on*,*pre*,*n*_*|V_ds_*_,*on*,*act*,1:*n*_) is used to sample the particles. Equation (9) can be approximated by calculating the average of these particles as follows:(10)$$\begin{array}{ll} E\left[f\left(V_{ds,on,pre,n}\right)\right] & \approx \frac{\frac{1}{N}\sum_{i=1}^{N}W_{n}\left(V_{ds,on,pre,n}^{\left(i\right)}\right)f\left(V_{ds,on,pre,n}^{\left(i\right)}\right)}{\frac{1}{N}\sum_{i=1}^{N}W_{n}\left(V_{ds,on,pre,n}^{\left(i\right)}\right)}\\ & =\overset{N}{\underset{i=1}{\sum }}{\tilde{W}}_{n}\left(V_{ds,on,pre,n}^{\left(i\right)}\right)f\left(V_{ds,on,pre,n}^{\left(i\right)}\right) \end{array}$$
where *N* is the number of particles, *W_n_* is the weight of the particle, and ${\tilde{W}}_{n}$ is the normalized value of *W_n_*, which can be calculated by:(11)$${\tilde{W}}_{n}\left(V_{ds,on,pre,n}^{\left(i\right)}\right)=\frac{W_{n}\left(V_{ds,on,pre,n}^{\left(i\right)}\right)}{\sum_{i=1}^{N}W_{n}\left(V_{ds,on,pre,n}^{\left(i\right)}\right)}$$

In order to improve the efficiency of calculation, the weight of particle *W_n_* is expressed recursively as follows:(12)$$\begin{array}{ll} W_{n}\left(V_{ds,on,pre,n}^{\left(i\right)}\right) & \propto W_{n-1}\left(V_{ds,on,pre,n}^{\left(i\right)}\right)\\ & \;\times \frac{p(V_{ds,on,act,n}|V_{ds,on,pre,n}^{\left(i\right)})p(V_{ds,on,pre,n}^{\left(i\right)}|V_{ds,on,pre,n-1}^{\left(i\right)})}{q(V_{ds,on,pre,n}^{\left(i\right)}|V_{ds,on,pre,n-1}^{\left(i\right)},V_{ds,on,act,n})} \end{array}$$

### 3.2. Extended Kalman Particle Filter

As mentioned above, the weight of many particles will be so small and can even be ignored after several iterations. Only a few particles have a large weight. This phenomenon is called particle degradation, which will lead to the degradation of estimation performance. A large number of computational resources will be consumed to deal with trivial particles. It not only causes the waste of resources but also affects the final results of estimation. In order to reduce the impact of particle degradation, three measures can be taken as follows:(1)Increase the number of particles

If the number of particles is large, the diversity of particles can be fully reflected and the degradation of particles can be slowed. However, it will inevitably increase the running time of the algorithm, which is not desirable in the application field of high real-time performance.

(2)Resampling

The resampling technique is introduced to avoid the problem of losing particle diversity, but it needs a recommended density distribution to sample particles.

(3)Choose a reasonable proposal density function

The Particle Filter algorithm must obtain a set of sample points from a reasonable posterior recommended density distribution. This set of sample points can cover the real state very well. If these assumptions are not met, the performance of the PF algorithm will degrade. Therefore, an optimal proposal density function must be found to sample points correctly. Then, the final quality of filtering can be guaranteed.

It is obvious that the combination of the second and third methods is the best choice. Resampling can sample the particles with high weight repeatedly and eliminate the particles with low weight. The number of particles will remain the same and each particle will have the same weight changed from *W_n_* to 1/*N* eventually. And the posterior probability function *p*(*V_ds_*_,*on*,*pre*,*n*_*|V_ds_*_,*on*,*act*,1:*n*_) can be expressed as follows,
(13)$$p(V_{ds,on,pre,n}|V_{ds,on,act,1:n})\approx \frac{1}{N}\overset{N}{\underset{i=1}{\sum }}\delta \left(V_{ds,on,pre,n}-V_{ds,on,pre,n}^{i}\right)$$

The optimal proposal distribution can transfer the sample points from the prior distribution region to the maximum likelihood region shown in Figure 8. Local linearization is an effective method to generate the proposal distribution. The Extended Kalman Filter (EKF) algorithm is a kind of method to achieve local linearization. It is a recursive minimum mean square error estimation method implemented by first-order Taylor expansion [30].

Therefore, the EKF algorithm is used to improve the PF algorithm in this paper. The improved algorithm is called Extended Kalman Particle Filter (EPF). The core of the EPF algorithm is that the EKF algorithm can guide the sampling of particles by calculating the mean and covariance of the particles in the sampling stage of the PF algorithm. It is used to produce a Gaussian recommended density distribution given by:(14)$$q(V_{ds,on,pre,n}^{\left(i\right)}|V_{ds,on,pre,n-1}^{\left(i\right)},V_{ds,on,act,n})=N\left({\bar{X}}_{n}^{\left(i\right)},P_{n}^{\left(i\right)}\right)$$
where ${\bar{X}}_{n}^{\left(i\right)}$ is the mean value and $P_{n}^{\left(i\right)}$ is variance of the *i*th particle, which can be calculated at time *n* − 1 by using the EKF algorithm and the latest observation information *V_ds_*_,*on*,*act*,*n*_ as follows:(15)$$q(V_{ds,on,pre,n}^{\left(i\right)}|V_{ds,on,pre,n-1}^{\left(i\right)},V_{ds,on,act,n})=N\left({\bar{X}}_{n}^{\left(i\right)},P_{n}^{\left(i\right)}\right)$$
(16)$$P_{n}^{\left(i\right)}=\left(I-K_{n}H_{n}^{\left(i\right)}\right)P_{n,pre}^{\left(i\right)}$$

Among which:(17)$$P_{n,pre}^{\left(i\right)}=F_{n}^{\left(i\right)}P_{n-1}^{\left(i\right)}{\left(F_{n}^{\left(i\right)}\right)}^{T}+Q_{n}$$
(18)$$K_{n}=P_{n,pre}^{\left(i\right)}{\left(H_{n}^{\left(i\right)}\right)}^{T}{\left(H_{n}^{\left(i\right)}P_{n,pre}^{\left(i\right)}{\left(H_{n}^{\left(i\right)}\right)}^{T}+R_{n}\right)}^{-1}$$
where *I* is the identity matrix, *F* and *H* are, respectively, the Jacobian matrices of the state transition function f and measurement function *h*, *Q* is the variance of process noise, *R* is the variance of measurement noise, and *K_n_* is the Kalman gain.

The algorithm flow of EPF-based RUL estimation is shown in Figure 9. First, the Equations (3) and (4) are built and parameters are initialized from the historical data of *V_ds_*_,*on*_. The EKF algorithm is used to provide an optimal proposal density distribution function for importance sampling of the PF algorithm. Then, the same proposal density distribution will be used to resample the set of particles. The measurement function *h* is updated by calculating the mean of the particles. The degenerate trajectory of *V_ds_*_,*on*_ is predicted by the modified function *h*. Finally, the RUL is estimated when the predicted value *V_ds_*_,*on*,*pre*,*n*_ is equal to the failure threshold *V_ds_*_,*on*,*th*_.

## 4. Remaining Useful Lifetime Prediction

### 4.1. Processing of Experimental Results and Selection of Degradation Model

The change of *V_ds_*_,*on*_ can represent the degradation of SiC MOSFETs. Since the input range of the analog-to-digital converter in the control board is 0 to 3.3 V, the *V_ds_*_,*on*_ is compressed within 3.3 V by a simple voltage divider circuit. Then, the moving average filter with a moving window width of 3 is used to process the *V_ds,on_* data to remove outliers. The degradation trend of *V_ds,on_* obtained by our power cycle test platform is shown in Figure 10. The failure threshold is 1.2 times the initial value of *V_ds,on_* [28]. It is obvious that the *V_ds,on_* is monotonically increasing.

The degradation model is very important for particle filter. Three different degradation empirical models are compared, which are represented as follows:(19)$$V_{ds,on}\left(t\right)=a_{i}\times \exp(b_{i}*t)+c_{i}\times \exp(d_{i}*t)$$
where *a_i_* = 2.342, *b_i_* = −0.0003897, *c_i_* = 0.1084, and *d_i_* = 0.01214;
(20)$$V_{ds,on}\left(t\right)=a_{g}\times \exp\left(-{\left(\frac{t-b_{g}}{c_{g}}\right)}^{2}\;\right)+d_{g}\times \exp\left(-{\left(\frac{t-e_{g}}{f_{g}}\right)}^{2}\;\right)$$
where *a_g_* = 2.861, *b_g_* = 368, *c_g_* = 153.3, *d_g_* = 2.637, *e_g_* = 818.4, and *f_g_* = 2906;
(21)$$V_{ds,on}\left(t\right)=a_{p}\times t^{3}+b_{p}\times t^{2}+c_{p}\times t+d_{p}$$
where *a_p_* = 1.158 × 10^−7^, *b_p_* = −4.969 × 10^−6^, *c_p_* = 0.001057, and *d_p_* = 2.443.

Figure 11 shows the prediction results of different degradation empirical models using PF. The root mean squared error (RMSE) is defined as follows:(22)$$\mathrm{RMSE}=\sqrt{\frac{1}{M}\overset{M}{\underset{i=1}{\sum }}{\left(y_{exp}^{i}-y_{pre}^{i}\right)}^{2}}$$
where *M* is the number of prediction data, *y_exp_* is the experimental data, and *y_pre_* is the prediction data. The RMSE of prediction results based on the degradation model represented by Equation (21) is significantly larger than the others. The prediction curve based on the degradation model represented by Equation (20) is too smooth to be affected by the change of *V_ds,on_*. The absolute error of prediction results based on the degradation model represented by Equation (20) is therefore larger than that based on the degradation model represented by Equation (19). As a result, the degradation empirical model represented by Equation (19) is selected in this work.

### 4.2. RUL Estimation Results

Since the measured value of *V_ds,on_* contains noise, the parameters of the Equation (19) also contain noise. Therefore, the state transition model of Equation (3) can be expressed as follows:(23)$$\left\{\begin{array}{c} a\left(n\right)=a\left(n-1\right)+w_{a}\left(n-1\right),w_{a}\left(n-1\right)∼N\left(0,\sigma_{a}\right)\\ b\left(n\right)=b\left(n-1\right)+w_{b}\left(n-1\right),w_{b}\left(n-1\right)∼N\left(0,\sigma_{b}\right)\\ c\left(n\right)=c\left(n-1\right)+w_{c}\left(n-1\right),w_{c}\left(n-1\right)∼N\left(0,\sigma_{c}\right)\\ d\left(n\right)=d\left(n-1\right)+w_{d}\left(n-1\right),w_{d}\left(n-1\right)∼N\left(0,\sigma_{d}\right) \end{array}\right.$$
where *σ_a_* = 0.001, *σ_b_* = 10^−6^, *σ_c_* = 0.001, *σ_d_* = 10^−4^.

Equation (23) represents state parameter information of *V_ds,on_*. The measurement function *h* in Equation (4) where *v_n_* = *N*(0,10^−3^) can be expressed by Equation (19). Then, Equation (23) is filtered by the EPF algorithm. In other words, the historical data are filtered to modify the parameters of the measurement function *h*. The subsequent change of *V_ds,on_* is predicted according to function *h* after updating.

The estimated *V_ds,on_* of different amounts of input data using PF and EPF are compared in Figure 12. The prediction error is defined as follows:(24)$$\mathrm{error}=\frac{\left|t_{f,act}-t_{f,pre}\right|}{t_{f,act}}\times 100\%$$
where *t_f,act_* and *t_f,pre_* are the real time and prediction time when *V_ds,on_* reaches the failure threshold voltage, respectively. The prediction errors with the EPF algorithm, respectively, are 18.1%, 4.8%, and 0%, with inputting 31%, 52%, and 78% of the aging data. The prediction errors with the PF algorithm are 31.3%, 14.5%, and 1.2%, respectively. The RUL estimation using the EPF algorithm is more accurate than that using the basic PF algorithm. Moreover, the EPF algorithm has a better prediction result even with few input data. The prediction error decreases with the increase of the amount of input data.

The estimated *RUL_t_* trajectories are shown in Figure 13. The *RUL_t_* is defined as follows:(25)$$RUL_{t}=t_{f}-t$$
where *t* is the current time, and *t_f_* is the prediction time when *V_ds,on_* reaches the failure threshold voltage. The RMSEs of the prediction trajectory are 255.8 for PF and 192.7 for EPF. The prediction trajectory of RUL using the EPF algorithm is closer to the real trajectory of RUL compared with that using PF algorithm, especially in the later life of SiC MOSFET. The remaining 10% of useful life can be used as a warning value to investigate the feasibility of an early warning for the system before device failure [15].

The absolute value of the prediction errors under different times are shown in Figure 14. As the *V_ds,on_* is closer to the failure threshold, the prediction results are more accurate. The prediction error approximates the form of an exponential function. The prediction errors of both methods are relatively large in the early time because the amount of data is small. Although the EPF algorithm improves the accuracy of prediction compared with the PF algorithm, the prediction error of RUL is still not satisfactory in the early time. The uncertainty caused by Gaussian distribution is the disadvantage of the EPF algorithm. The interval between line aa’ and line bb’ is selected as the optimal prediction period of EPF algorithm which the prediction error is below 20%. The optimal prediction period of EPF algorithm is obviously longer than that of PF algorithm, which makes it a promising candidate of prediction algorithms for SiC MOSFETs.

The RUL prediction errors of different kinds of prediction methods for SiC MOSFETs are compared in Table 4. It is obvious that the RUL prediction method using the EPF algorithm based on *V_ds,on_* degradation model in this paper has the smallest prediction error, which makes it a promising candidate for RUL prediction.

## 5. Conclusions

This paper focuses on the estimation of remaining useful life using the EPF algorithm based on the *V_ds,on_* degradation model for power SiC MOSFETs. This EPF algorithm overcomes the problem of particle degradation in the PF algorithm, which improves the prediction accuracy. The *V_ds,on_* obtained by a newly designed power cycle test platform is used as the failure precursor. Three different *V_ds,on_* degradation empirical models are also discussed. The experimental results show that the RUL prediction error is 11.5% based on EPF algorithm reduced from 20.5% based on PF algorithm with 40% data input. The life prediction accuracy is therefore improved by about 10%.

## Figures and Tables

**Figure 1 micromachines-14-00836-f001:**
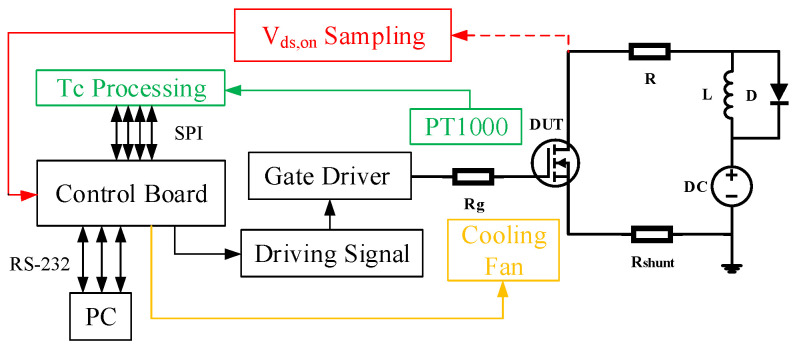
Schematic diagram of the experimental setup.

**Figure 2 micromachines-14-00836-f002:**
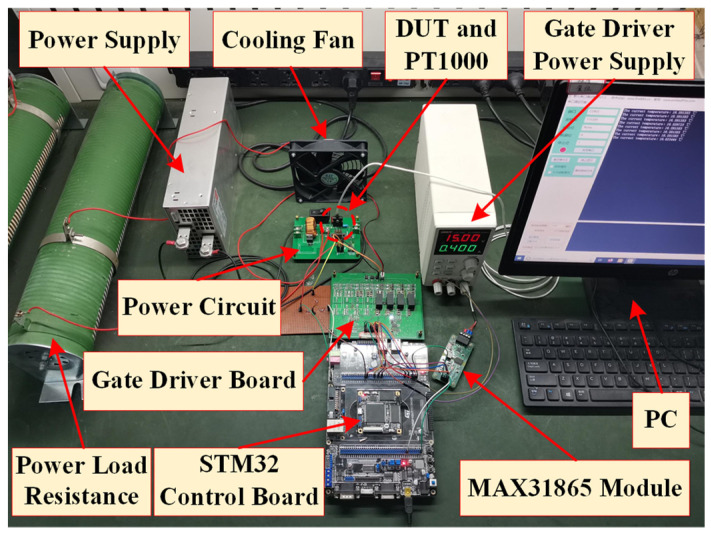
The physical photo of the experimental platform.

**Figure 3 micromachines-14-00836-f003:**
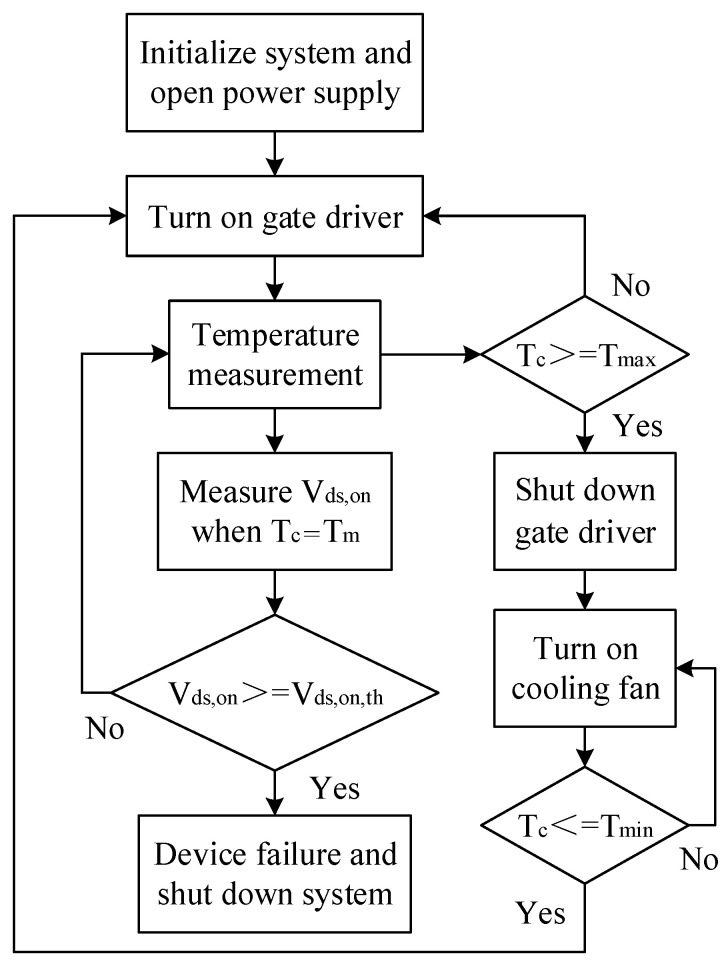
The program flow of the experiment.

**Figure 4 micromachines-14-00836-f004:**
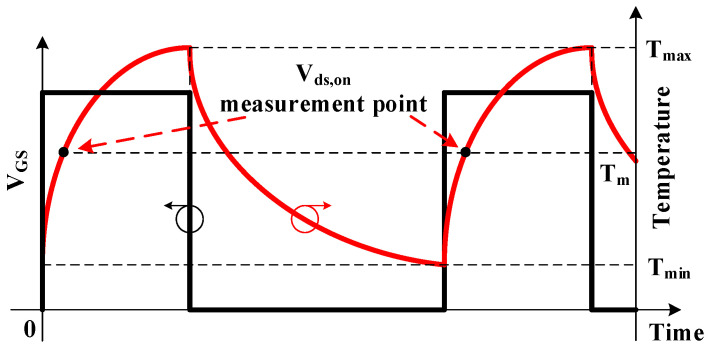
Temperature swing during the power cycling test.

**Figure 5 micromachines-14-00836-f005:**
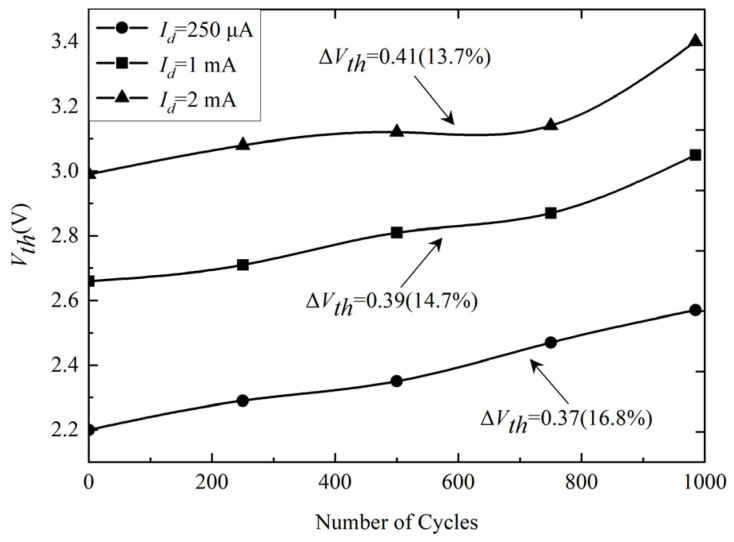
The variation in *V_th_* for different drain currents *I_d_*.

**Figure 6 micromachines-14-00836-f006:**
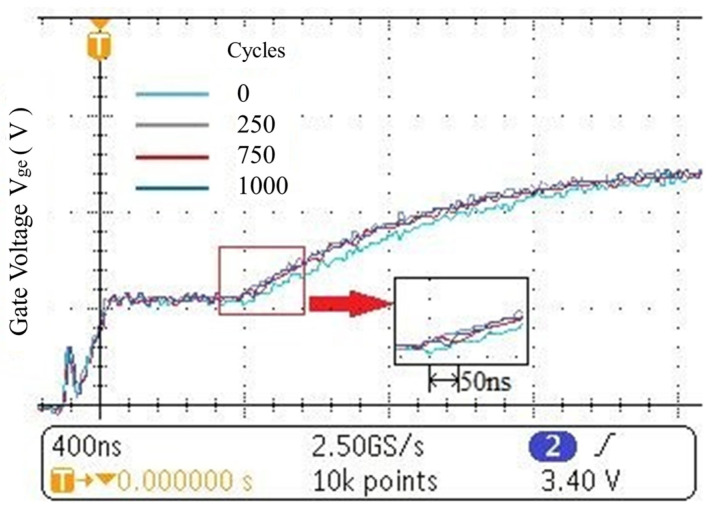
The gate voltage waveforms during turn-on at different numbers of power cycles.

**Figure 7 micromachines-14-00836-f007:**
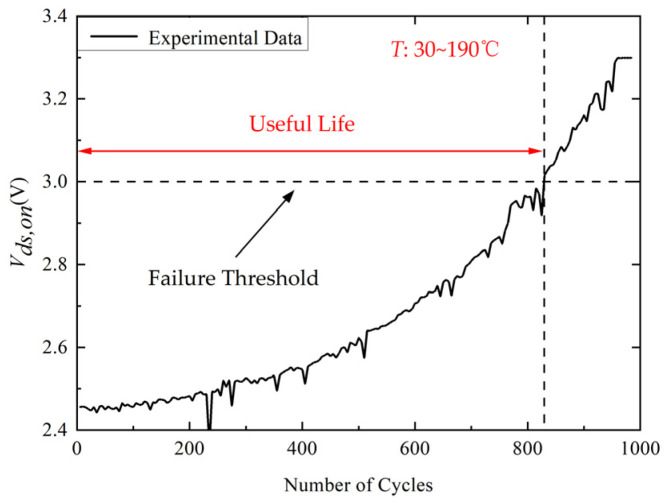
The experimental data of *V_ds,on_* (Δ*T* = 160 °C).

**Figure 8 micromachines-14-00836-f008:**
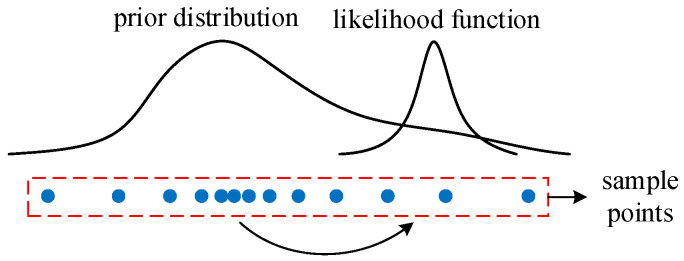
Transfer of sample sets.

**Figure 9 micromachines-14-00836-f009:**
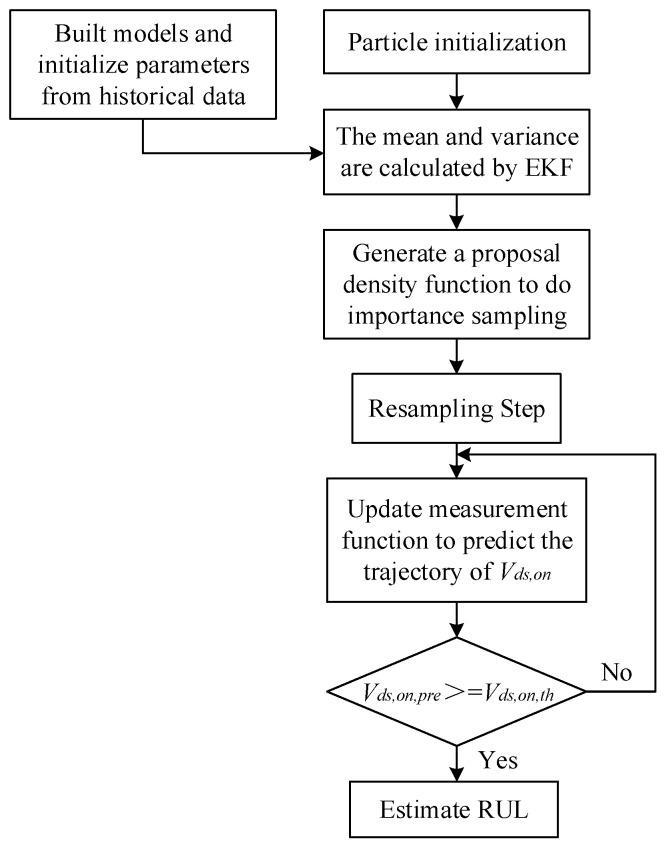
An EPF-based RUL estimation algorithm flow.

**Figure 10 micromachines-14-00836-f010:**
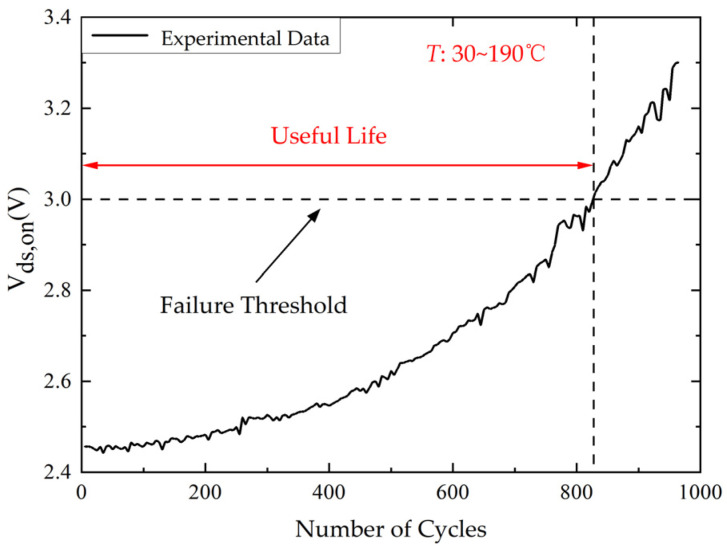
The experimental data of *V_ds,on_* processed by moving average filter (Δ*T* = 160 °C).

**Figure 11 micromachines-14-00836-f011:**
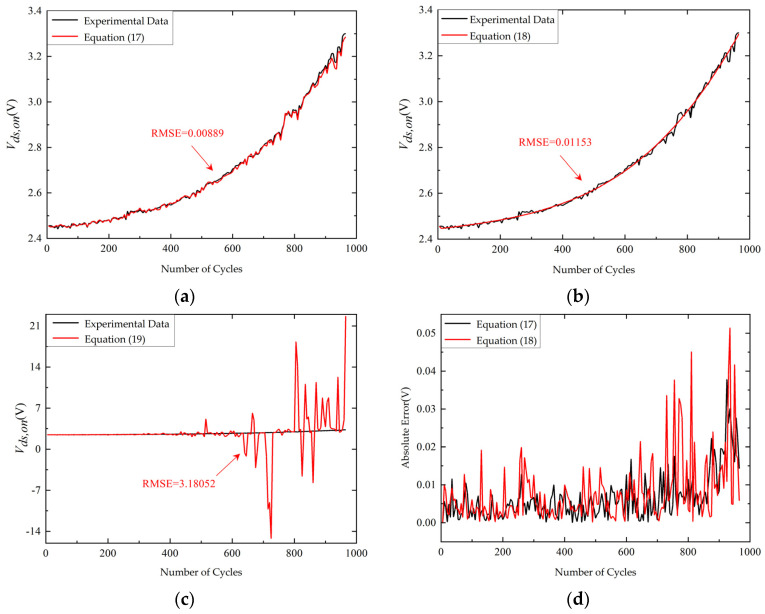
The prediction results of different degradation empirical models using PF. (**a**) The prediction curve based on the degradation model represented by Equation (19); (**b**) the prediction curve based on the degradation model represented by Equation (20); (**c**) the prediction curve based on the degradation model represented by Equation (21); (**d**) the absolute error of prediction curve based on the degradation model represented by Equation (19) and the prediction curve based on the degradation model represented by Equation (20).

**Figure 12 micromachines-14-00836-f012:**
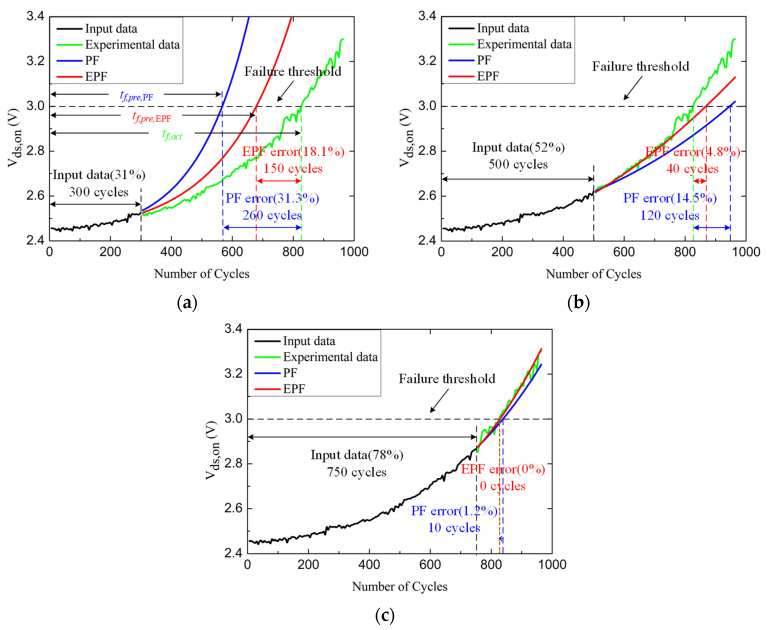
The prediction results for different amounts of input data. (**a**) 31% input data. (**b**) 52% input data. (**c**) 78% input data.

**Figure 13 micromachines-14-00836-f013:**
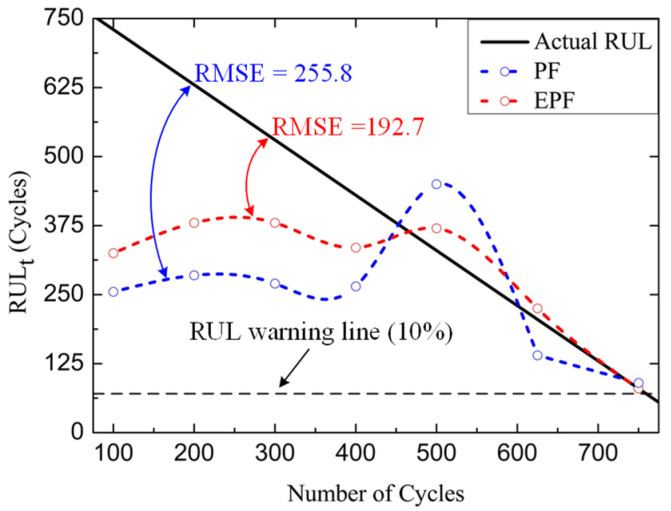
The estimated RUL trajectories based on the EPF and PF.

**Figure 14 micromachines-14-00836-f014:**
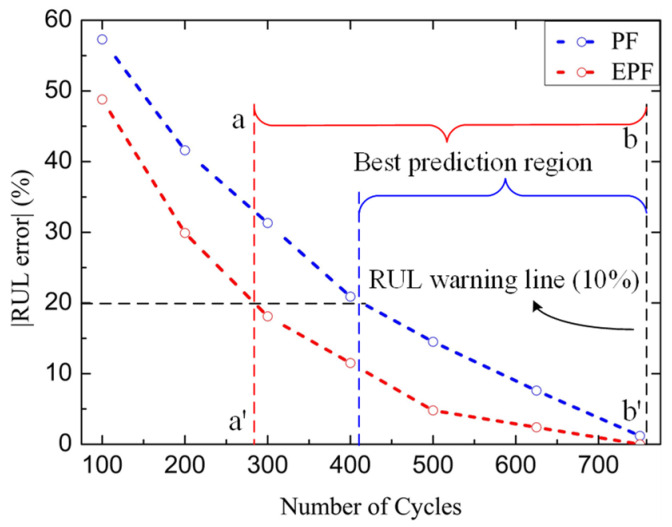
Comparison of RUL prediction errors between the EPF and PF.

**Table 1 micromachines-14-00836-t001:** Some main parameters of Si and SiC materials.

Material	Si	4H-SiC
Forbidden band width (eV, 300 K)	1.1	3.3
Critical electric field (MV/cm)	0.3	2.8
Electron mobility (cm^2^·V^−1^·s^−1^)	1400	900
Hole mobility (cm^2^·V^−1^·s^−1^)	500	100
Thermal conductivity (W·cm^−1^·K^−1^)	1.5	4.9

**Table 2 micromachines-14-00836-t002:** The main parameters of experimental setup.

Power load resistor, *R*	5 Ω
Power supply, DC	40 V
Minimum case temperature, *T_min_*	30 °C
Maximum case temperature, *T_max_*	190 °C
Average case temperature, *T_m_*	110 °C
Gate voltage, *V_GS_*	15 V

**Table 3 micromachines-14-00836-t003:** Comparison of different failure precursors.

Δ*V_th_*		Δ*V_ds,on_*		Δ*t_GP_*	
mV	%	mV	%	ns	%
0.37	16.8	0.86	35.2	90	24

**Table 4 micromachines-14-00836-t004:** Comparison of different kinds of RUL prediction methods.

Prediction Method	EPF			PF			Stochastic Degradation Model [22]		
Failure Precursor	*V_ds,on_*						*C_iss_*, *C_oss_*, *C_res_*, *V_sd_*, *V_gs,th_*, *R_ds,on_*		
Amounts of input data	40%	55%	70%	40%	55%	70%	40%	55%	70%
Computation time (ms)	39	51	68	31	26	33	-	-	-
Prediction error	11.5%	4.2%	1.2%	20.9%	12.8%	4.4%	15.0%	5.52%	22.5%

## Data Availability

Not applicable.

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
