# Peer review of "Remaining Useful Lifetime Prediction Based on Extended Kalman Particle Filter for Power SiC MOSFETs"

_micromachines, 2023, doi:10.3390/mi14040836_

Round 1

Reviewer 1 Report

This paper has presented a RUL prediction research work based on particle filter, the appllication is for power electronics. 

1) particle filter has been widely ued in the RUL prediction work, it can be used independently or combained with other techniques such as signal processing as presented in  Early-Stage end-of-Life prediction of lithium-Ion battery using empirical mode decomposition and particle filter. Proceedings of the Institution of Mechanical Engineers, Part A: Journal of Power and Energy (2023): 09576509231153907. Therefore, what's your main contribution in this paper? Could you please point out your main contributions clearly in the introduction part ? 

2) more recent research work should be cited and discussed in your introduction including model-based or data-driven RUL prediction techniques such as A degradation empirical-model-free battery end-of-life prediction framework based on gaussian process regression and Kalman filter. Have you campared your propsed method with orther data-driven techniques? or have you compared your research work based on different degradation empirical models ? because the degradation model is very important for particle filter.

3) Could you please add more information in the abstract and conclusion part ? for example, the prediction accuracy etc. 

Author Response

This paper has presented a RUL prediction research work based on particle filter, the application is for power electronics. 

Point 1: Particle filter has been widely used in the RUL prediction work, it can be used independently or combined with other techniques such as signal processing as presented in Early-Stage end-of-Life prediction of lithium-Ion battery using empirical mode decomposition and particle filter. Proceedings of the Institution of Mechanical Engineers, Part A: Journal of Power and Energy (2023): 09576509231153907. Therefore, what's your main contribution in this paper? Could you please point out your main contributions clearly in the introduction part? 

Response 1: There are two contributions in this paper. Firstly, a new power cycling test platform with on-line monitoring of on-state voltage is proposed in this paper. Secondly, three different Vds,on degradation empirical models for particle filter are discussed and a remaining useful life estimation method using EPF based on Vds,on degradation empirical models for SiC MOSFETs is proposed for the first time. And the main contributions have been pointed out in the introduction part from line 81 to line 88.

Point 2: More recent research work should be cited and discussed in your introduction including model-based or data-driven RUL prediction techniques such as a degradation empirical-model-free battery end-of-life prediction framework based on Gaussian process regression and Kalman filter. Have you compared your proposed method with other data-driven techniques? Or have you compared your research work based on different degradation empirical models? Because the degradation model is very important for particle filter.

Response 2: Three recent research work include the one mentioned above have been cited and discussed in the introduction part from line 39 to line 52. Some prediction methods for SiC MOSFET are compared in Table 3 in the paper. However, there are too few prediction methods for SiC MOSFET based on data-driven techniques. Because the Failure precursor of Si devices and SiC devices have different sensitivity to device aging, there is no comparison between RUL prediction accuracy of Si devices and that of SiC devices. And the comparison of three different degradation empirical models has been added in part 4.1 from line 265 to line 284.

Point 3: Could you please add more information in the abstract and conclusion part? For example, the prediction accuracy etc. 

Response 3: The information of improved prediction accuracy has been added in the abstract in line 17 and conclusion part in line 340.

Reviewer 2 Report

EPF is used for life prediction for SiC MOSFETs.  EPF is a very common algorithm. It has been used in many works. So, what's the core highlight of your paper? Why does the author not conduct actual testing? The sensor accuracy is of vital in the experiment. How do you ensure accuracy? How do you avoid noise?

Author Response

EPF is used for life prediction for SiC MOSFETs.  EPF is a very common algorithm. It has been used in many works. So, what's the core highlight of your paper? Why does the author not conduct actual testing? The sensor accuracy is of vital in the experiment. How do you ensure accuracy? How do you avoid noise?

Point 1: What’s the core highlight of your paper?

Response 1: There are two highlights in this paper. Firstly, a new power cycling test platform with on-line monitoring of on-state voltage is proposed in this paper. Secondly, three different Vds,on degradation empirical models for particle filter are discussed and a remaining useful life estimation method using EPF based on Vds,on degradation empirical models for SiC MOSFETs is proposed for the first time.

Point 2: Why does the author not conduct actual testing?

Response 2: An accelerated degradation experiment for SiC MOSFETs is implemented based on our proposed power cycling test platform to reveal the variation trend of Vds,on. And the remaining useful life estimation method using EPF for SiC MOSFETs is verified by the degradation trend of Vds,on obtained from the accelerated degradation experiment.

Point 3: How do you ensure accuracy? How do you avoid noise?

Response 3: The advanced sensor chips are used in our proposed power cycling test platform to collect Vds,on data. Then the Vds,on data is processed by the moving average filter to remove outliers.

Round 2

Reviewer 1 Report

The questions have been answered, the paper can be accpeted. 

Author Response

Point 1: the questions have been answered, the paper can be accpeted.  

Response 1: Thanks very much!

Reviewer 2 Report

The author answered some of my questions. The following revisions are required before the paper is published.

1. How to ensure that MOSFET is one fault and there are no other agings.

2. Computation time of the proposed methods also need to be calculated and compared.

3. The authors proposed the PF and EKF integrated model, but the authors dont give any parameters of the model. The parameter values are very important in the model.

4. Many integrated methods of data driven have been proposed for prediction and the authors miss these. Such as VMD, PF and GPR integrated (10.1109/TVT.2021.3138959), LSTM and BLS integrated (10.1016/J.EST.2022.104901), PSO and BLS integrated (10.3389/FENRG.2022.1013800). The author needs to discuss these.

5. There are too fewer discussions and the discussions lacks mechanism reason.

Author Response

Point 1: How to ensure that MOSFET is one fault and there are no other agings.

Response 1: Firstly, the power cycling test implemented in this paper mainly causes package-level failures which mainly including solder layer fatigue and bond wire failure. And the bond wire failure is the most common failure mode of package-level failures. Relevant statements have been added in lines 109 to line 120. Secondly, data-driven RUL prediction techniques are based on the variation in failure precursors which is one of the concerns in this paper. A discussion of failure precursors selection have been added in lines 161 to line 205.

Point 2: Computation time of the proposed methods also need to be calculated and compared.

Response 2: the computation time of the proposed methods have been add and compared in Table 4.

Point 3: The authors proposed the PF and EKF integrated model, but the authors don’t give any parameters of the model. The parameter values are very important in the model.

Response 3: Some parameter values have been given in line 347 to line 349. And Most of the remaining parameter values are calculated by the degraded data.

Point 4: Many integrated methods of data driven have been proposed for prediction and the authors miss these. Such as VMD, PF and GPR integrated (10.1109/TVT.2021.3138959), LSTM and BLS integrated (10.1016/J.EST.2022.104901), PSO and BLS integrated (10.3389/FENRG.2022.1013800). The author needs to discuss these.

Response 4: the research work mentioned above have been cited and discussed in the introduction part in line 51 to line 64

Point 5: There are too fewer discussions and the discussions lacks mechanism reason.

Response 5: the discussion of three different failure precursors have been added in line 161 to line 205 including three pictures and a table.
